Subject Area:
developmental biology/cellular biology/genetics

Keywords:
niche ageing, *mastermind*, reactive oxygen species, DE-cadherin, Hedgehog, *Drosophila* oogenesis

Author for correspondence:
Acaimo González-Reyes
e-mail: agonrey@upo.es

# *mastermind* regulates niche ageing independently of the *Notch* pathway in the *Drosophila* ovary

María Lobo-Pecellín, Miriam Marín-Menguiano and Acaimo González-Reyes

Centro Andaluz de Biología del Desarrollo, CSIC/Universidad Pablo de Olavide/JA, Carretera de Utrera km 1, 41013 Sevilla, Spain

 AG-R, 0000-0003-3048-0968

Proper stem cell activity in tissues ensures the correct balance between proliferation and differentiation, thus allowing tissue homeostasis and repair. The *Drosophila* ovary develops well-defined niches that contain on average 2–4 germline stem cells (GSCs), whose maintenance depends on systemic signals and local factors. A known player in the decline of tissue homeostasis is ageing, which correlates with the waning of resident stem cell populations. In *Drosophila*, ovaries from old females contain fewer GSCs than those from young flies. We isolated niche cells of aged ovaries, performed a transcriptomic analysis and identified *mastermind (mam)* as a factor for *Drosophila* ovarian niche functionality during ageing. We show that *mam* is upregulated in aged niche cells and that we can induce premature GSC loss by overexpressing *mam* in otherwise young niche cells. High *mam* levels in niche cells induce reduced *Hedgehog* amounts, a decrease in cadherin levels and a likely increase in reactive oxygen species, three scenarios known to provoke GSC loss. Mam is a canonical co-activator of the Notch pathway in many *Drosophila* tissues. However, we present evidence to support a Notch-independent role for *mam* in the ovarian germline niche.

## 1. Introduction

Stem cells are essential for tissue homeostasis and are involved in the regeneration of damaged organs. Stem cells respond to signals of different types and realms of action so that their activity is regulated by environmental factors and by local and systemic signals, including ageing. Indeed, the decline in regenerative capacity characteristic of aged organs is linked to the impaired activity of their hosted stem cells [1]. The fact that ageing impacts tissue-specific stem cells, and thus organ function, demands the identification of the precise molecular mechanisms behind stem cell obsolescence.

The gradual loss of tissue homeostasis during ageing is a consequence of local modifications in the stem cells themselves and in their supporting microenvironment or niche. Thus, ageing stem cells show changes in the expression of genes involved in the preservation of genomic integrity, and in transcriptional and epigenetic regulation; they can display both declined responsiveness to extrinsic signals and impaired adhesion to support cells or extracellular matrix; and they can increase intracellular concentrations of reactive oxygen species (ROS), accumulate DNA damage and show slower proliferation and deficient self-renewing divisions [2–8]. Similarly, the support cells that form part of the ageing niches undergo changes that eventually affect their homed populations of tissue-specific stem cells. For example, experiments involving the parabiosis of heterochronic mice have identified molecular and cellular mechanisms behind the age-related dysfunction of muscles that map to the aged niches [9]. Other studies in *Drosophila* have shown that niche cell numbers can decrease with time, which in turn affects the pool of stem cells hosted within [10]. In addition, ageing niche cells

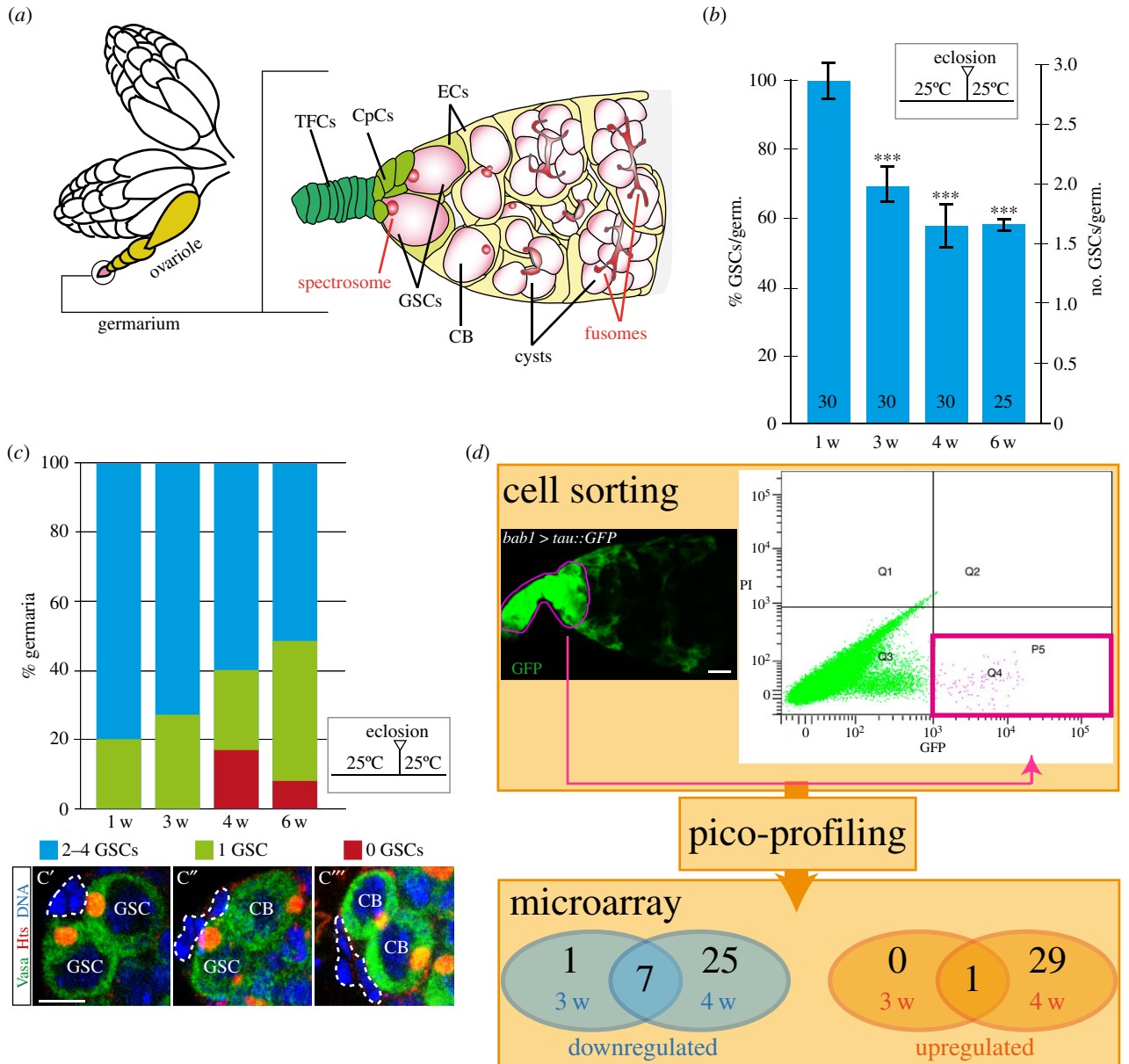

**Figure 1.** Transcriptomic analysis of niche support cells during ageing. (*a*) Schematic of an ovary pair with detail of a germarium (adapted from [13]). In the apical part of the germarium, terminal filament cells (TFCs), cap cells (CpCs) and the escort cells (ECs) that surround the germline stem cells (GSCs) form the germline niche. Also shown are the differentiating cystoblasts (CB) and cysts, the GSC and CB spectrosomes and the cyst fusomes. (*b*) Quantification of GSCs/germarium (*Y*-axis, right) and the percentage of GSC reduction (*Y*-axis, left) in young (one week) compared with old (three, four and six weeks) females. (*c*) Percentage of germaria from the dataset in (*b*) containing 0, 1 or 2–4 GSCs. (*c′*–*c′″*) Niches containing two GSCs, one GSC and one CB, or two CBs stained with a DNA dye and with antibodies to Hts (red) and to Vasa (green) to visualize spectrosomes and germline cytoplasm, respectively. Dashed lines indicate CpC clusters. (*d*) Pattern of *bab1-Gal4* expression in TFCs and CpCs (purple line, high levels of expression) and in escort cells (lower levels), as shown by the Tau::GFP reporter. Work flow for the transcriptomic study: sorting of bright (green fluorescent protein (GFP) signal, *X*-axis), alive (propidium iodide (PI), *Y*-axis) niche cells from *bab1 > tau::GFP* ovaries was followed by mRNA pico-profiling and microarray analysis. Transcriptome comparisons of one-week-old cells with three- or four-week-old samples yielded a number of downregulated or upregulated genes. In this and the following figures, boxes in the graphs indicate the treatment of the flies prior to dissection. In this and the rest of the figures, only *p*-values of two-tailed *t*-tests considered statistically significant between control and experimental samples are indicated (*$p \leq 0.05$, **$p \leq 0.005$, ***$p \leq 0.0005$). Numbers in bars represent germaria analysed. Scale bars, 5 µm.

may reduce the production of niche signals and of adhesion molecules and increase ROS contents, all of which compromise the maintenance of stem cell populations [2,6]. Therefore, as a result of ageing, stem cell populations shrink in numbers or become functionally impaired with time.

The *Drosophila* ovary is composed of functional egg-producing tubes called ovarioles. Each ovariole contains tissue-resident stem cells homed in an experimentally accessible niche that is susceptible to genetic manipulations and microscopic analysis. The coordinated activity of the stem cell populations in the niche fuels the generation of new egg

gametes during oogenesis. Located in the germarium, a tapered structure found at the anterior tip of the ovariole, this niche is composed of extracellular matrix and a few meso-dermal cell types that offer support to 2–4 germline stem cells (GSCs) and to a larger population of somatic stem cells termed follicle stem cells (FSCs) [11,12]. The GSC niche includes terminal filament cells (TFCs), cap cells (CpCs) and a special group of anterior escort cells (ECs; figure 1*a*). Both CpCs and the anterior ECs are in contact with GSCs and are necessary to maintain the pool of undifferentiated GSCs through adhesion and signalling events [14–16]. GSCs can be recognized by

their position adjacent to the CpCs and by the presence of a membranous structure called the spectrosome in their cytoplasm. Upon mitosis, a GSC can give rise to another GSC and to a differentiating cystoblast (CB), which divides synchronously to produce two-, four-, eight- and 16-cell cysts. Ovarian niche ageing in *Drosophila* conveys a significant reduction in the number of GSCs hosted in the niche [6,17,18] (figure 1*a–c*).

To try to identify new genes required in niche cells and involved in GSC depletion with ageing, we embarked upon a transcriptomic analysis of maturing niche support cells in *Drosophila* and found a discrete number of candidate genes whose transcription was regulated with age. We focused our efforts on the *mastermind (mam)* gene, a known cofactor involved in the canonical transduction of the Notch signal and in the transcription of target genes [19]. Mam forms a ternary complex with an intracellular processed form of the Notch receptor and Suppressor of Hairless (CSL in humans). The *Notch* pathway is required for the architecture of the GSC niche, where it controls the number of CpCs and thus niche size, and for the maintenance of the correct number of GSCs during adulthood [17,20–25]. We found, however, that *mam* participates in ovarian niche ageing independent of the *Notch* pathway.

## 2. Results

### 2.1. A transcriptomic analysis identifies *mastermind* as a candidate gene involved in ovarian stem cell niche ageing

In order to identify genes involved in the ageing of stem cell microenvironments, we first confirmed that ovaries from aged, mated *Drosophila* females contain germaria with fewer GSCs than controls [6,17,18]. In addition, the proportion of niches devoid of GSCs in four- or six-week-old adults was significantly higher than that in one-week-old controls (figure 1*b,c*). Next, we used fluorescence-activated cell sorting (FACS) to isolate green fluorescent protein (GFP)-labelled TFCs and CpCs from one-, three- and four-week-old females and performed a transcriptomic analysis using DNA microarrays (approx. 400 cells/array). We identified 30 upregulated and 32 downregulated genes in four-week-old females compared with one-week-old controls (false discovery rate < 0.05; linear fold change >2; figure 1*d*; electronic supplementary material, figure S1 and table S1; because of technical limitations, we could not isolate enough niche cells from six-week-old females to perform a similar analysis). Among the upregulated genes, *mastermind (mam)* was expressed 3.58- and 2.47-fold higher in three- and four-week-old TFCs and CpCs, respectively, than in one-week-old controls. This was further confirmed by measuring directly *mam* RNA levels corresponding to the long and short isoforms using digital polymerase chain reaction (PCR) in isolated one-, three- and four-week-old niche cells (figure 2*a*). Considering that *mam* was the only gene upregulated in the three-week-old sample and that it maintained a higher expression in older cells, we decided to analyse in deeper detail the role of *mam* during ovarian niche ageing.

### 2.2. High *mam* levels in niche cells induce GSC loss

To validate a possible role for *mam* in aged niches, we first looked at *mam* expression, making use of an enhancer of the *mam* gene

(GMR28A08; [26]) to drive the expression of a GFP reporter and found that the enhancer is active in TFCs, CpCs and ECs (figure 2*b*). Second, and considering that an increase of *mam* levels in niche cells correlated with a decrease in GSC numbers, we specifically knocked down *mam* mRNA amounts in adult niche cells to check if lower *mam* levels could prevent the drop in GSC numbers typical of ageing niches. We used the *bab1-Gal4* driver combined with the *Gal-80^{ts}* to overexpress after eclosion an RNA interference (RNAi) version of the *mam* gene. *bab1-Gal4* is expressed at high levels in adult TFCs and CpCs; it is also expressed, albeit weakly, in ECs [27]. We raised flies at 22°C and 25°C for four weeks to achieve different degrees of RNAi and found that, while experimental flies grown at 22°C behaved like controls in terms of GSCs/germarium (controls, 2.21 GSCs/germarium on average, number of germaria analysed ($n$) = 38; *bab1 > mam RNAi*, 2.23 GSCs/germarium, $n$ = 30) and of Hedgehog and *DE*-cadherin levels (see below), raising the flies at 25°C induced GSC loss (controls, 1.74 GSCs/germarium, $n$ = 31; *bab1 > mam RNAi*, 1.21 GSCs/germarium, $n$ = 29; electronic supplementary material, figure S2A, S2B). We interpret this result as indicating that a certain activity of *mam* is required for GSC maintenance (in accordance with its known role in Notch signalling; see below in the Notch section).

Third, to determine if the mere increase of *mam* levels could mimic niche ageing and thus GSC loss, we used a similar approach to overexpress a full-length version of the *mam* gene (*mam*^Long; see below) in the somatic cells of the adult niche (hereafter referred to as the *bab1 > mam* condition). Overexpressing *mam* for one or two weeks at consistent levels in the ovarian niche of freshly eclosed females gradually induced a significant GSC loss (one-week-old controls, 2.52 GSCs/germarium, $n$ = 79; one-week-old *bab1 > mam*, 2.20 GSCs/germarium, $n$ = 114; two-week-old controls, 2.45 GSCs/germarium, $n$ = 65; two-week-old *bab1 > mam*, 1.78 GSCs/germarium, $n$ = 66; figure 2*c*). In addition, we also observed that the proportion of niches containing one or zero GSCs rose from 15% to 30% in females overexpressing *mam* for two weeks (figure 2*d*; see also controls in electronic supplementary material, figure S2C). These changes in GSC behaviour were not a consequence of modifying CpC numbers or fate, as determined by Lamin-C staining of control and experimental niches (figure 6*c,d*).

The *mam* gene encodes two different protein products, Mam-^Long and Mam^Short. The latter lacks some C-terminal sequences essential for Notch signalling and it is known to act as a dominant-negative version that interferes with Mam^Long and that represses the Notch pathway in certain cell types [28]. Since aged niche cells showed higher mRNA levels of both *mam* isoforms (figure 2*a*), we wished to check whether *mam*^Short affected the GSC niche. We thus overexpressed *mam*^Short for one week and found that GSC numbers did not change compared with controls (*bab1 > mam*^Short; electronic supplementary material, figure S2D). This result suggests that *mam*^Short does not have a critical function in this context. Finally, we determined that one-week overexpression of full-length *mam* in somatic cells of the ovary with the *traffic jam-Gal4* driver did not affect *mam*^Short levels (electronic supplementary material, figure S2E), strongly suggesting that inducing high levels of *mam*^Long in the ovary represents a true *mam* gain-of-function situation.

Because *bab1-Gal4* is expressed at high levels in adult TFCs and CpCs and in ECs [27], we wished to determine if *mam* overexpression in TFCs/CpCs or ECs would provoke GSC depletion. To this end, we overexpressed *mam* for two weeks with the *hedgehog-Gal4* line, expressed in adult TFCs and

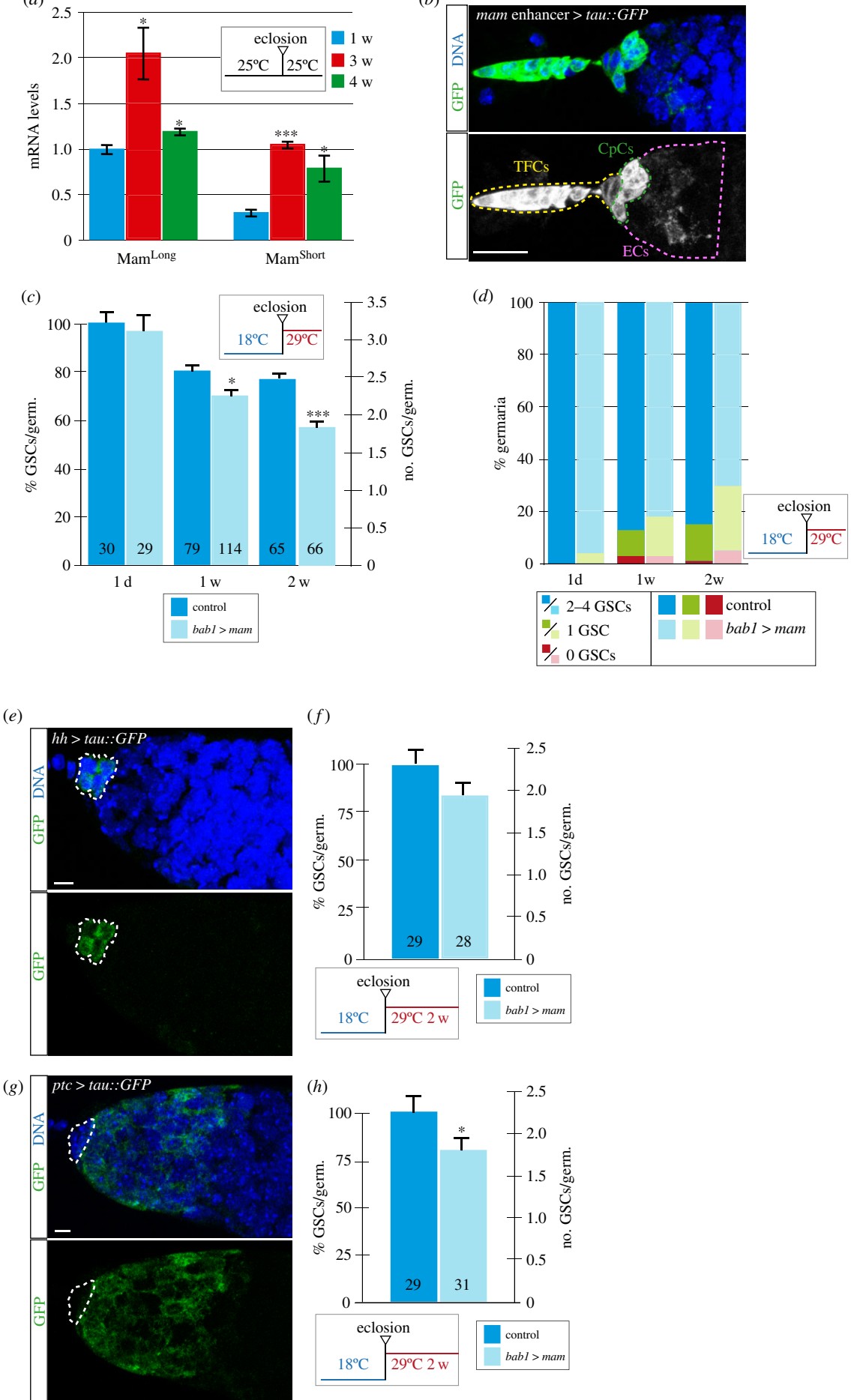

**Figure 2.** (*Caption opposite.*)

royalsocietypublishing.org/journal/rsob Open Biol. 9: 190127

**Figure 2.** (*Opposite.*) High *mam* levels in niche cells induce GSC loss. (*a*) Droplet Digital PCR data showing that *mam* RNA levels are increased in aged, isolated CpCs and TFCs. (*b*) Expression of the *tau::GFP* reporter driven by the *mam* GMR28A08 enhancer in TFCs, CpCs and ECs. (*c*) Quantification of GSCs/germarium after over-expressing *mam* for 1 day, one week or two weeks after eclosion using the *bab1-Gal4* driver in niche cells (*bab1 > mam*). (*d*) Percentage of germaria from the dataset in (*c*) with 0, 1 or 2–4 GSCs in *bab1 > mam* niches of different age. (*e*) Pattern of expression of the *hedgehog* (*hh*)-*Gal4* line using the Tau::GFP reporter after keeping the adults for one week at 29°C. (*f*) *hh > mam* flies kept for two weeks at permissive temperature (29°C) do not show significant changes in GSC numbers compared with controls. (*g*) Pattern of expression of the *patched* (*ptc*)-*Gal4* line using the Tau::GFP reporter after keeping the adults for one week at 29°C. (*h*) *ptc > mam* flies kept for two weeks at 29°C contain significantly lower GSC numbers. Darker columns correspond to controls; lighter columns indicate experimental data. Control genotypes in (*c,f,h*) contain the *UAS-mam* chromosome in the absence of a Gal4 driver. See also electronic supplementary material, figure S2 for additional controls. Numbers in bars represent germaria analysed. Error bars are shown for the different samples. Scale bars, 5 µm.

CpCs. Although the tendency was to find fewer GSCs in these experimental niches, there was no significant GSC loss compared with controls (two-week-old controls, 2.28 GSCs/germarium on average, $n = 29$; two-week-old *hh > mam*, 1.93 GSCs/germarium on average, $n = 28$; figure 2*e,f*; see also controls in electronic supplementary material, figure S2F). Next, we drove full-length *mam* expression in adult ECs for two weeks using the *patched-Gal4* line and found that the number of GSCs present in experimental niches, even though significantly lower in *ptc > mam* (two-week-old controls, 2.24 GSCs/germarium on average, $n = 30$; two-week-old *ptc > mam*, 1.79 GSCs/germarium, $n = 29$; figure 2*g,h*; see electronic supplementary material, figure S2F for additional controls), was not as reduced as in *bab1 > mam*. These data thus suggest that high levels of *mam* are necessary in TFCs/CpCs and ECs to account for the total GSC loss typical of aged niches. In all, from the above results, we conclude that *mam* levels increase with time in the niche, and that this increase is sufficient to induce GSC loss.

## 2.3. *mam* overexpression decreases *D*E-cadherin levels in niche cells

Next, we sought to determine the reason(s) for GSC loss upon induction of high *mam* levels. We dismissed cell death as one-week-old *bab1 > mam* germaria did not increase expression of the Dcp-1 caspase in GSCs (electronic supplementary material, figure S3). Considering the role of GSC–niche cell adhesion in stem cell maintenance [29], we decided to analyse cadherin-mediated adhesion in control and experimental germaria. It has been previously reported that ageing affects adhesion in GSC niches in both *Drosophila* males and females, and that this is due to a decrease in cadherin levels [2,6]. To test if the GSC loss characteristic of *mam*-overexpressing niches could be due to a reduction in cadherin levels, we studied the distribution of *D*E-cadherin in CpC-CpC and CpC-GSC interfaces of aged and *bab1 > mam* niches. As a reporter for *D*E-cadherin expression, we used a *D*E-cadherin::GFP fusion protein under the control of the endogenous regulatory sequences of the gene [30]. When analysing one-week-old versus six-week-old germaria, we found a significant decrease in *D*E-cadherin::GFP levels, with aged niches containing on average *D*E-cadherin levels of 74% of the controls (figure 3*a,b*). We then examined the effect on *D*E-cadherin::GFP levels of overexpressing *mam* for one or two weeks in experimental germaria and found that, while one-week-old overexpressing germaria behaved like controls, *D*E-cadherin::GFP levels were significantly reduced after two weeks of overexpression (figure 3*c,d*; experimental germaria contained on average 86% *D*E-cadherin::GFP levels of controls). To further confirm this result,

we measured endogenous, untagged *D*E-cadherin protein levels by immunostaining and found that they were reduced by nearly 50% after two weeks of *mam* overexpression (electronic supplementary material, figure S4A,B). Together, the above data demonstrate that *mam* overexpression in young niches recapitulates the adherens junction defects typical of older niches. They also suggest that high levels of *mam* are sufficient to induce premature niche ageing and the concomitant loss of GSCs by impairing cadherin levels and thus niche cell–GSC adhesion. Finally, to test if the reduction in *D*E-cadherin levels upon ectopic expression of *mam* is functionally relevant, we checked whether *D*E-cadherin overexpression could rescue the GSC loss characteristic of *bab1 > mam* germaria. We found that a two-week induction of high *D*E-cadherin amounts in *bab1 > mam* niche cells did not rescue the decrease in GSC contents characteristic of high *mam* levels (electronic supplementary material, figure S4C).

## 2.4. Aged niches show defective *D*E-cadherin trafficking

Once we had confirmed that the overexpression of *mam* induced a precocious decrease in *D*E-cadherin levels in niche cells, we determined if membrane *D*E-cadherin turnover was also affected by age and/or *mam* overexpression. We set out to measure the dynamics of protein replacement in CpC cyto-plasmic membranes by performing FRAP (fluorescence recovery after photobleaching) experiments to look at the recovery of *D*E-cadherin::GFP signal in one-week-old versus four-week-old females (the *D*E-cadherin::GFP signal in females older than four weeks was too low to allow consistent measurements). In young samples from *D*E-cadherin::GFP females, bleached CpC membranes reached a maximum of fluorescence recovery close to 30% of the pre-bleached signal 30 min after photobleaching, which was in the same range as previously published data involving *D*E-cadherin::GFP FRAP analyses in embryos [31]. By contrast, four-week-old cells subjected to the same experimental conditions recovered less than 10% of fluorescence 30 min after photobleaching, suggesting that ageing jeopardizes trafficking and replacement of cell-surface *D*E-cadherin (figure 4*a,b*). Next, we analysed the membrane-bound fusion protein Resil::GFP as a turnover control, and noticed that one-week-old samples recovered 40% of the pre-bleached signal 30 min after photobleaching, while four-week-old TFCs and CpCs recovered around 20% (figure 4*c,d*). Thus, the mere ageing of niche cells can affect membrane protein replacement dynamics. It remains to be determined if the small difference in fluorescence recovery observed on *D*E-cadherin::GFP trafficking in aged niches compared with that on Resil::GFP implies that ageing impairs *D*E-cadherin recycling to a larger extent than other membrane proteins.

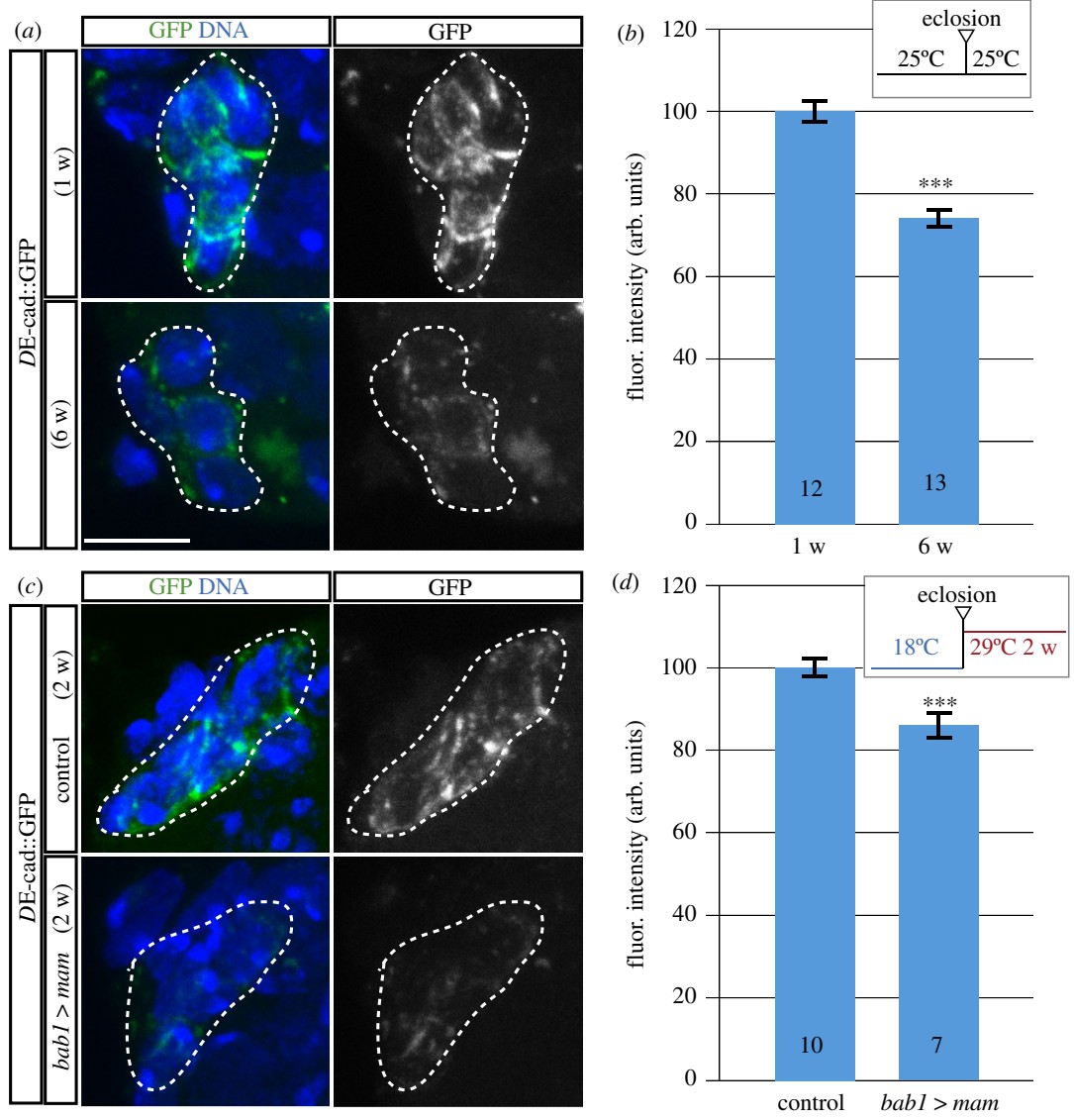

**Figure 3.** High *mam* levels in niche cells reduce *D*E-cadherin amounts. (*a*) Z-projections of germaria from one-week-old and six-week-old flies stained to visualize DNA and the fusion protein *D*E-cadherin::GFP. (*b*) Quantification of *D*E-cadherin::GFP levels shows that they decrease with age in niche cells. (*c*) Z-projections of germaria from control and experimental flies overexpressing *mam* for two weeks stained to visualize DNA and the fusion protein *D*E-cadherin::GFP. (*d*) Quantification of *D*E-cadherin::GFP levels to show their decrease in niches overexpressing *mam* for two weeks. Numbers in bars represent germaria analysed. Data plotted in (*b*,*d*) were obtained from at least 15 measurements/germarium. Dashed lines outline CpC clusters. Scale bars, 5 µm. In this and the following figures, fluor. intensity (arb. units) stands for fluorescence intensity (arbitrary units).

The Rab11 GTPase has been implicated in *D*E-cadherin recycling from internalized endosomes [32]. Nuclear fallout (Nuf) is an adaptor protein that interacts with Rab11 to control the recycling endosome [33]. In order to test if niche cell ageing affected membrane protein trafficking by changing Rab11-Nuf levels, we determined whether Rab11 and Nuf protein amounts in CpCs varied during ageing conditions. We compared one- and six-week-old germaria and observed a significant reduction in both Rab11 and Nuf contents in the aged cells (figure 4*e*,*f*; electronic supplementary material, figure S5C,D).

To address the role of high *mam* levels in *D*E-cadherin recycling, we first performed FRAP on *D*E-cadherin::GFP TFCs and CpCs overexpressing *mam* for one week but could not detect any differences in fluorescence recovery with control flies (electronic supplementary material, figure S5A,B), indicating that *D*E-cadherin::GFP dynamics are not changed in one-week-old *bab1 > mam* flies. This result is in line with our previous observation that ovarian niches in these experimental females show

normal levels of *D*E-cadherin::GFP. Unfortunately (as shown in figure 3*c*,*d*), overexpression of *mam* for two weeks reduces *D*E-cadherin::GFP levels to the extent that we could not perform FRAP analyses reliably. Second, we observed that after two weeks of *mam* overexpression Rab11 levels were decreased, suggesting that high *mam* quantities affected membrane protein recycling. However, we failed to detect changes in Nuf amounts upon *mam* overexpression for two weeks (figure 4*g*–*h*; electronic supplementary material, figure S5E,F). In all, our results demonstrate that physiologically aged niche cells recycle membrane proteins (such as *D*E-cadherin or Resil) less efficiently than younger ones. Moreover, considering the reduction in *D*E-cadherin, Rab11 and Nuf levels in aged niches (and of *D*E-cadherin and Rab11 in high *mam* conditions), our results may suggest that ageing favours the sorting of internalized *D*E-cadherin (and probably many other membrane proteins such as Resil) towards the lysosomal degradation pathway rather than towards the Rab11-mediated recycling [34,35], and that *mam* overexpression induces a similar scenario.

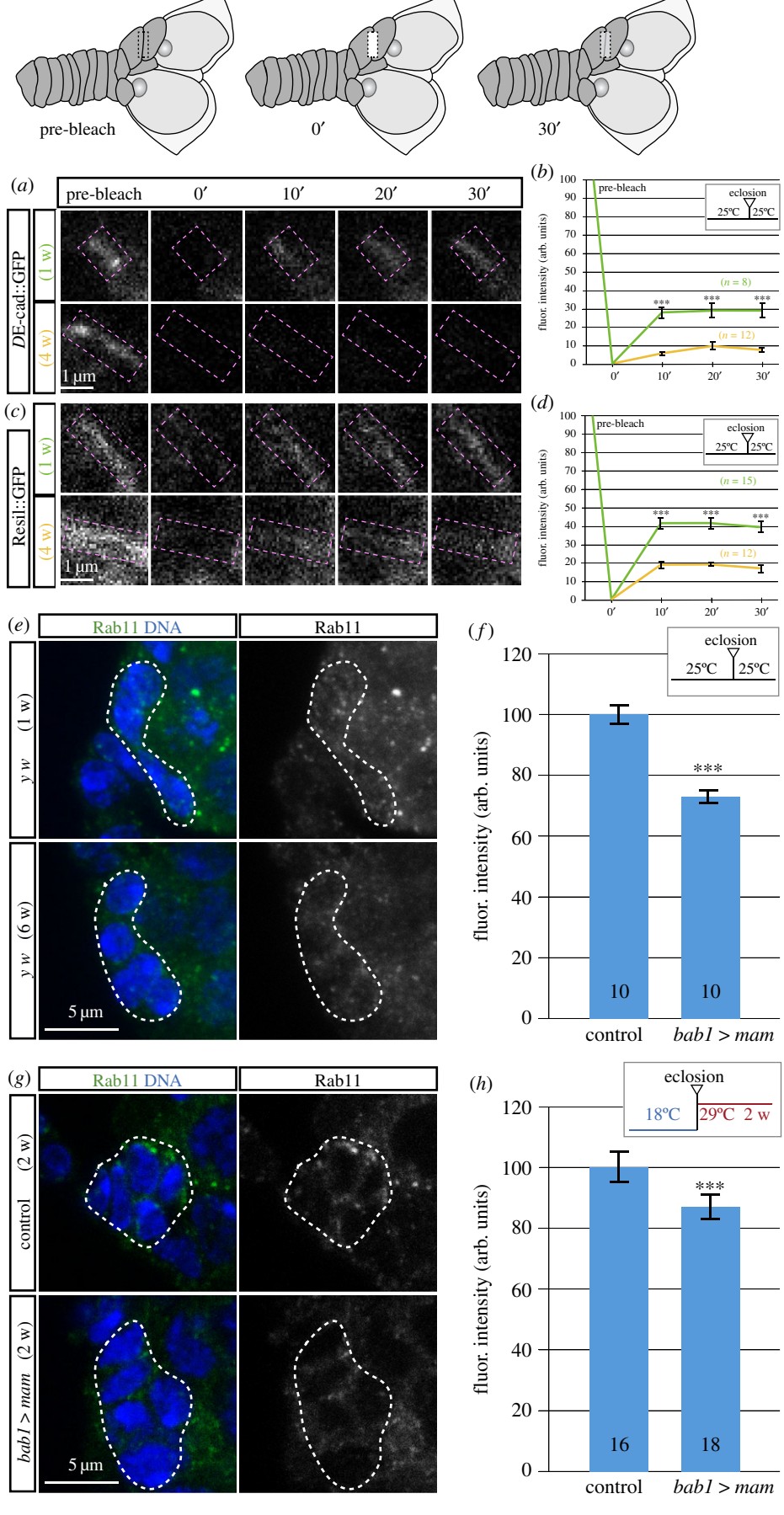

**Figure 4.** (*Caption opposite.*)

## 2.5. *mam* expression regulates Hedgehog levels in the ovarian niche

Our previous results indicate that GSC loss in *bab1 > mam* germaria is not due to apoptosis. In addition, in the course of our

studies, we found germline cells in contact with *bab1 > mam* CpCs that possessed spectrosomes typical of CBs or fusomes characteristic of two- and 4-cell cysts—a situation never encountered in controls and indicative of GSC loss by differentiation. Thus, because *mam* overexpression did not affect

**Figure 4.** (*Opposite.*) *DE*-cadherin recycling and Rab11 levels change during ageing. *mam* overexpression affects Rab11 amounts. The dynamics of *DE*-cadherin::GFP recycling were analysed using FRAP, as shown in the upper drawings. (*a*) Images showing FRAP in one-week-old and four-week-old CpCs. (*b*) Young niches recover 30% of fluorescence 30 min after photobleaching whereas old niches barely recover 10%. (*c*) FRAP in one-week-old and four-week-old CpCs expressing the membrane-bound Resil::GFP fusion. (*d*) Young niches recover 40% of fluorescence 30 min after photobleaching whereas old niches recover 20%. (*e*) Z-projection of germaria from young (one week) and aged (six weeks) flies stained with antibodies to Rab11 (green) and with the DNA dye Hoechst (blue). (*f*) Rab11 levels in niche cells decrease with age. (*g*) Z-projection of germaria from *bab1 > mam* flies kept for two weeks at 29°C stained with antibodies to Rab11 (green) and with the DNA dye Hoechst (blue). (*h*) *mam* overexpression reduces Rab11 levels in experimental CpCs. Data plotted in (*f,h*) were obtained from at least 15 measurements/germarium. Purple dashed boxes in (*a,c*) demarcate the photobleached regions; dashed lines in (*e,g*) outline CpC clusters. Numbers in bars represent germaria analysed. Scale bars, 5 μm. *n* = number of germaria analysed. Pre-bleach levels are considered 100% intensity and the post-bleach signal is arbitrarily considered 0%. Later time points are referenced to the post-bleach values.

royalsocietypublishing.org/journal/rsob    *Open Biol.* **9**: 190127

niche cell numbers (figure 6*c,d*), the decrease of GSC numbers in ovarian niches expressing high levels of *mam* could be due to flawed niche signalling that, in turn, induced GSC differentiation. In this context, it has been shown that aged niches are less effective in transducing the BMP pathway or in keeping constant levels of the Hedgehog (Hh) protein, two events that act normally in the niche to maintain a proliferative GSC pool [6,15,18,36,37]. We thus decided to analyse whether *mam* overexpression had an effect on the signalling properties of the GSC niche. First, we quantified the activity of the BMP pathway by scoring the intensity of the BMP activity reporter phosphorylated Mothers against *dpp* (pMad) in aged GSCs containing normal-looking spectrosomes. In contrast to published results [18], we could not detect a significant decrease in pMad levels in six-week-old GSCs (electronic supplementary material, figure S6A,B), suggesting that, at least under our conditions, BMP signalling might not be affected by ageing. We found the same scenario when looking at *bab1 > mam* niches, which did not show a decrease in pMad levels even after a two-week exposure to high *mam* levels (electronic supplementary material, figure S6C,D). Second, we looked at the Hedgehog pathway. The Hh ligand is expressed in TFCs and CpCs, where it acts to maintain the GSC pool in adult ovaries [15,36]. Interestingly, the analysis of six-week-old niches provided a clear difference in Hh signal when compared with one-week-old controls (figure 5*a,b*). This finding is in agreement with a previous report showing that aged TFCs/CpCs exhibited lower activation of a *hh-lacZ* enhancer trap [18]. Similarly, niches overexpressing *mam* (*bab1 > mam*, two weeks at 29°C) also displayed significantly lower levels of Hh amounts (figure 5*c,d*). To test if ageing and/or *mam* overexpression affected not only Hh levels in CpCs but also Hh signalling in the target cells, we used an antibody to Patched that gives a clean, specific staining in the wing disc and we scored Patched amounts in the anterior half of the germarium. The Patched (Ptc) receptor is a known target of Hh signalling and its transcription is a direct readout of the pathway [38,39]. In control germaria, Ptc is found in puncta in the areas that the separate the different germline cysts and that are occupied by the ECs, similar to the expression reported for a *ptc-lacZ* construct [15]. CpCs did not show any specific visible signal. This pattern of expression recapitulates that described for a reporter construct known to mimic *hh*-pathway signalling, Ptc-pelican-GFP [40]. However, and in spite of the reduced Hh amounts under both experimental conditions, we found that ECs of aged and *bab1 > mam* germaria kept for two weeks at 29°C contained Ptc levels similar to their respective controls (figure 5*e–g*).

From the above experiments, we conclude that both ageing and *mam* overexpression can induce a decrease in Hh levels in

niche cells. Surprisingly, we also found that the activity of the pathway—at least as indicated by Ptc levels—seemed to be maintained in the above conditions, suggesting that other modulators apart from the Hh ligand may act during its signalling in the germarium. One such modulator reported to act in other tissues such as imaginal discs is *dachshund* [41]; however, in our experiments, Dachshund was not expressed in the germarium. Thus, the mechanism(s) responsible for the enhanced Hh-independent signalling in aged ovarian niches remains to be identified. In this context, we also tested whether the overexpression of a functional version of the Hh protein, Hh::GFP [42], could rescue the loss of GSC numbers induced by high *mam* levels and found that it did not (electronic supplementary material, figure S6E).

## 2.6. Increased antioxidant activity can rescue premature ageing due to *mam* overexpression

ROS have been suggested to cause ageing, and previous results indicate that reduction of ROS in either GSCs or their niche improves GSC proliferation and lifespan, pointing towards an age-dependent increase in ROS levels in niches and stem cells [6]. In order to determine if high *mam* levels in the niche could induce premature ROS-dependent GSC loss, we tested whether reducing oxidative species in *bab1 > mam* niches could improve GSC maintenance and its effect on Hh and *DE*-cad amounts. To this end, we increased the amounts of the well-known antioxidant enzyme superoxide dismutase (SOD1) in the cytoplasm of niche cells and determined that SOD overexpression rescued the GSC loss characteristic of *bab > mam* niches (two-week-old controls, 2.91 GSCs/germarium, *n* = 35; two-week-old *bab1 > mam + GFP*, 2.05 GSCs/germarium, *n* = 37; two-week-old *bab1 > mam + SOD1*, 2.93 GSCs/germarium, *n* = 40; figure 6*a*). We also confirmed that Hh and *DE*-cad levels in experimental flies were similar to those in controls (electronic supplementary material, figure S7A). Since overexpression of SOD1 on its own in niche cells does not affect GSC numbers (electronic supplementary material, figure S7B), we thus propose that *mam* overexpression in niche cells increases ROS levels and that this contributes to the reduction in GSC numbers found in *bab1 > mam* niches.

## 2.7. High *mam* levels induce GSC loss in a *Notch*-independent manner

Mam is a transcriptional co-activator that supports the activity of the Notch receptor by forming a ternary complex with the DNA-binding protein CSL (Su(H) in *Drosophila*) and the Notch intracellular domain, NICD or $N^{intra}$ [19,43]. Thus, Mam is proposed to act as an essential component of

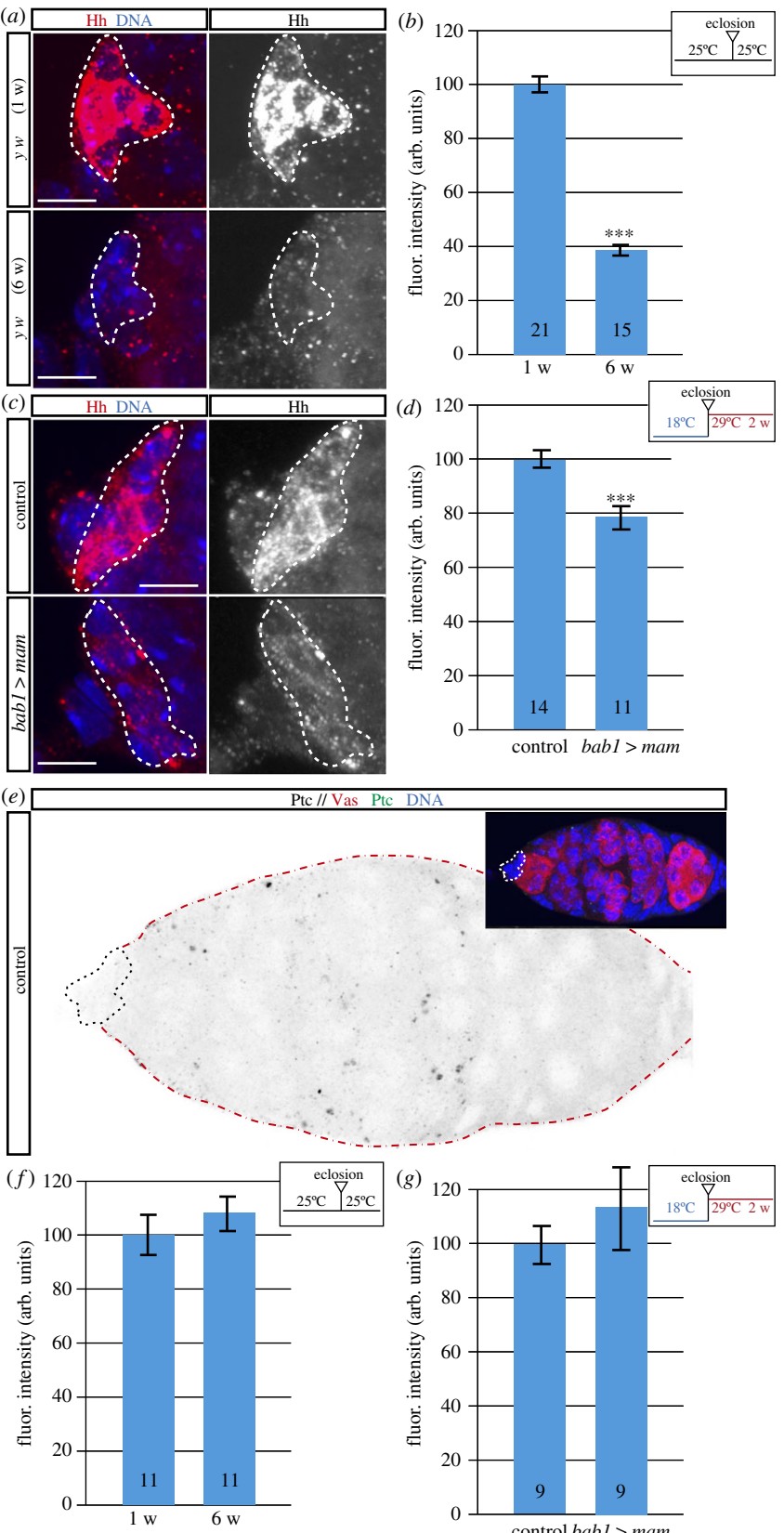

**Figure 5.** *mam* overexpression impairs niche signalling. (*a*) Z-projections of germaria from 1-week-old and 6-week-old flies stained to visualize the Hedgehog (Hh) protein (red) and DNA (blue). (*b*) Quantification of Hh levels in CpCs shows that Hh amounts decrease with age. (*c*,*d*) Similar experiments to (*a*,*b*) but comparing the amounts of Hh in control and experimental flies overexpressing *mam* for two weeks. Numbers in bars represent germaria analysed. (*e*) Germarium from a one-week-old control stained with an anti-Patched antibody to visualize the distribution of the Patched (Ptc) protein. The inset shows the same germarium but highlighting the germline cells (positive for Vasa; red) and DNA contents (Hoechst; blue). (*f*) Graph showing Ptc levels in germaria from one- or six-week-old females. (*g*) Graph showing Ptc levels in germaria from control or *bab1 > mam* females. Data plotted in (*b*,*d*) were obtained from at least 15 measurements/germarium. Black dashed lines outline CpC clusters. Red dashed lines outline the germarium. Scale bars, 5 μm.

canonical Notch signalling in *Drosophila* by increasing the dwell time of Su(H) complexes on target promoters [44]. Interestingly for our work, it has been reported that Notch signalling is required for niche maintenance in adult ovaries and that the activity of the pathway varies depending on the cell type. Thus, while Notch signalling increases with time in

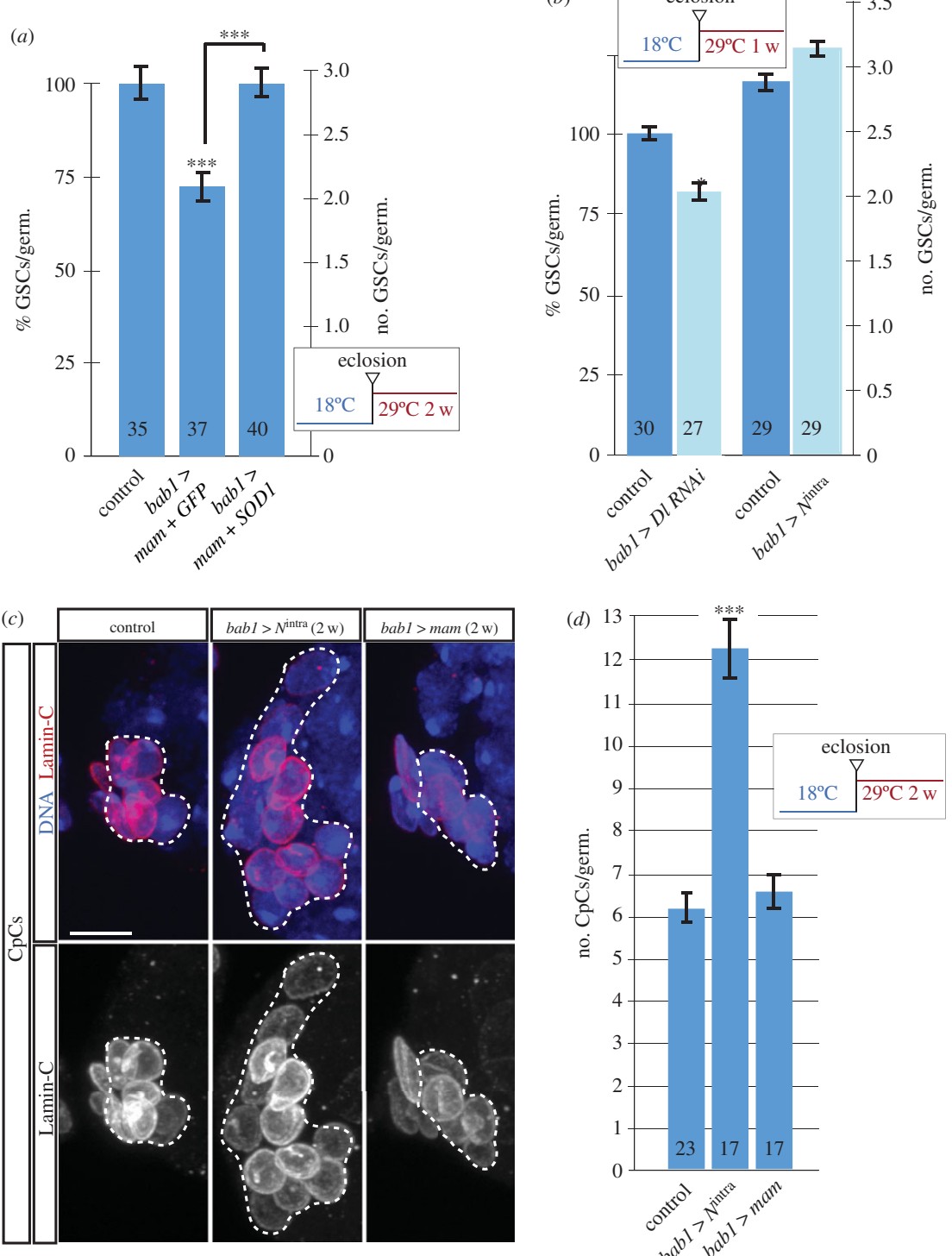

**Figure 6.** *mam* overexpression induces ROS-dependent GSC loss. High *mam* levels act independently of the Notch pathway. (*a*) Increasing the levels of the SOD enzyme rescues the GSC loss due to *mam* overexpression. (*b*) Graph showing that *bab1 > Dl RNAi* induces GSC loss, whereas *bab1 > N*^intra does not affect total GSC quantities. (*c*) Z-projections of control and experimental germaria overexpressing *N*^intra or *mam* stained to label TFC and CpC nuclear membranes (Lamin-C; red) and DNA (blue). (*d*) Quantification of CpCs/germarium to show that overexpressing *N*^intra increases CpC numbers, whereas high *mam* levels do not. Dashed lines outline CpC clusters. Numbers in bars represent germaria analysed. Scale bars, 5 µm.

GSCs, it decreases in seven-week-old niche cells [17,21], suggesting that low levels of Notch pathway activity in niche cells correlate with the GSC loss characteristic of aged niches, in line with our finding that *bab1 > mam RNAi* females grown at 25°C contained fewer GSCs than their controls (electronic supplementary material, figure S2A). However, we have found that *mam* levels augment at least during the first four weeks in aged niche cells and that an increase in *mam* mRNA in somatic niche cells induces GSC loss, results

that are seemingly contradictory to Mam's role as a cofactor of the Notch pathway. Hence, we set out to investigate whether the observed higher *mam* levels in three- and four-week-old niche cells may act Notch-independently. First, we confirmed that loss of function of the Notch pathway (by reducing the amount of the Delta ligand in niche cells) reduced GSC numbers (one-week-old controls, 2.53 GSCs/germarium, *n* = 30; one-week-old *bab1 > Dl RNAi*, 2.13 GSCs/germarium, *n* = 27; figure 6*b*), corroborating that

royalsocietypublishing.org/journal/rsob   Open Biol. 9: 190127

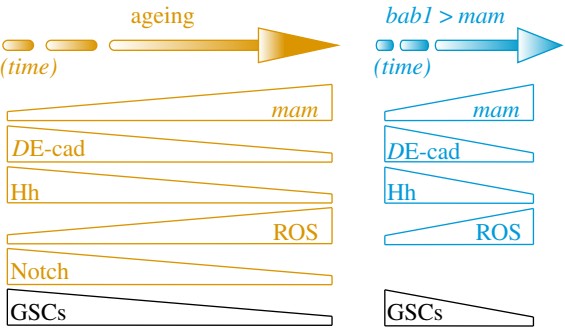

**Figure 7.** A working model. In our current view, niche cell ageing induces high *mam* expression, which results in higher ROS contents, lower overall *DE*-cadherin levels and a decrease in Hedgehog protein. All these factors, together with inefficient *DE*-cadherin recycling in aged niche cells, contribute to the GSC loss typical of old niches. *mam* overexpression mimics some of these characteristics. This role of *mam* is independent of its canonical function in the transmission of the Notch signal, which declines with time in the niche cells.

adult germaria depended on Notch signalling in niche cells for GSC maintenance [17,21]. Second, we showed that increasing Notch signalling in niche cells (by overexpressing $N^{intra}$ in niche cells) did not induce GSC loss, a finding in sharp contrast to the decline in GSC numbers typical of *mam* overexpression (one-week-old controls, 2.98 GSCs/germarium, $n = 29$; one-week-old *bab1 > $N^{intra}$*, 3.16 GSCs/germarium, $n = 29$; figure 6b). To prove that the Notch pathway was activated in *bab1 > $N^{intra}$*, we determined the number of CpCs present in control and experimental niches. It has been previously shown that ectopic activation of the Notch pathway in niche cells induces extra CpCs [23]. In our experimental conditions, two weeks of $N^{intra}$ overexpression nearly duplicated the number of CpCs present in experimental germaria (*bab1 > $N^{intra}$*). By contrast, ectopic expression of *mam* for two weeks (*bab1 > mam*) did not affect CpC numbers (figure 6c,d). Thus, the fact that increasing *mam* mRNA levels in niche cells induced GSC loss (whereas ectopic Notch signalling did not) and the observation that overexpression of $N^{intra}$ augmented CpC numbers (in contrast to *mam* overexpression, which did not) led us to conclude that the induction of high *mam* mRNA levels during niche ageing affects the GSC pool in a Notch-independent pathway and that high Mam levels in aged niche cells might interact with partners different from the CSL/NICD complex figure 7.

## 3. Discussion

### 3.1. *mam*, a novel factor in stem cell niche ageing

The ageing of tissues correlates with a decline in regenerative capacity and in homeostasis, linking the activity of tissue-specific stem cells to age-related organ dysfunction. Several factors had been reported to cause stem cell waning during ageing, mainly increased ROS production, decreased signalling efficiency and compromised adhesion to niche support cells, all of which can jeopardize stem cell maintenance within the niche. These 'ageing' factors can be intrinsic to the stem cells, or affect their surrounding microenvironments, which also contribute to impaired tissue activity in aged organisms. We have identified *mam* as a novel factor involved in the ageing of a particular stem cell microenvironment, the ovarian niche of *Drosophila melanogaster*.

Our transcriptomic analysis reveals a discrete number of genes whose expression is either upregulated or downregulated in niche support cells aged for three or four weeks, which represents around the half-life of a laboratory fly. The fact that the identified genes in the three-week-old samples are only a small fraction of those in four-week-old cells (and *mam* is a good example of this, as it is the only gene upregulated in three-week-old samples whereas in four-week-old niches there are 30+ upregulated genes), and that all but one candidate maintain their differential expression one week later, suggests that ageing is a gradual process in which few genes implicated in seemingly unrelated biological processes contribute to the progressive loss of stem cells from the niche.

*mam* is upregulated in three- and four-week-old niche cells, albeit with a smaller increase in the latter. Interestingly, when artificially induced, *mam* overexpression in niche cells is enough to bring about stem cell loss in young tissues in a manner that recapitulates niche ageing, as it can induce *DE*-cadherin loss; it decreases the levels of a key niche signalling molecule (Hh); and it most likely increases ROS amounts in niche support cells. The link between these important events in stem cell maintenance is currently missing, as it relates to why higher *mam* levels would induce oxidative stress most likely by increasing ROS quantities. We can only speculate that overexpression of *mam* in otherwise young niches might induce an accumulation of ROS-dependent damage in the cell, which ultimately results in impaired signalling, recycling and adhesion within the niche. Perhaps high *mam* amounts accelerate the cell's metabolism, thus increasing the generation of free radicals to the extent that the cell cannot cope with them efficiently. Alternatively, or in addition, *mam* overexpression might impede the normal activity of the antioxidant defence system present in niche cells. Whatever the mechanism, this places the physiological increase in *mam* levels in a preeminent position in the ageing of the ovarian niche, at least during the first half of adult life. Furthermore, the fact that 'young' stem cells can be lost from artificially aged niches indicates that support niche cells are fundamental to regulating stem cell waning from old niches and that the physiological condition of the microenvironment may determine the maintenance of stem cells independently of stem cell age. This observation emphasizes the constraints that niche physiology poses on stem cell-based therapies.

### 3.2. An unconventional function for high *mam* levels independent of Notch in aged niches

The Notch signalling pathway operates in a myriad of evolutionarily conserved developmental and disease processes. It ensures the release of NICD upon ligand binding. Partnered with the DNA-binding protein CSL and the co-activator Mam, the NICD/CSL/Mam complex regulates expression of target genes. Modulation of this process involves several factors, including ligand and receptor expression and levels, regulation of the processing steps required to release NICD and crosstalk with other signalling pathways [45,46]. There are, however, examples in which CSL or Mam can act independently of their interaction with Notch to induce different cellular phenotypes [47]. For instance, CSL can engage the pancreas transcription factor PTF-1 to regulate exocrine pancreatic genes [47,48]. In *Drosophila*, Su(H) acts as a Notch-independent activator to allow proper bristle formation in the adult notum

[49]. As for Mam, it is known that it can interact with other factors not included in Notch signalling, such as MEF2C, p53, β-catenin and NF-κB [50–53]. In this context and since Mam functions as a co-activator in numerous cellular processes, Mam family proteins have been proposed to mediate cross-talk among multiple signalling pathways involving Notch signalling or independent of it [54]. In the case of *Drosophila*, several studies have also reported a non-canonical role for Mam in different systems such as the central nervous system, where *Notch* and *mam* mutants display distinct phenotypes [55], or the wing and eye imaginal discs, where Mam regulates Wnt pathway activity independently of Notch through the activity of specific, additional factors [56]. Interestingly, considering that Wnt signalling controls the redox state of germline and somatic cells of the germarium [57], *mam* may provide a link between *mam*, Wnt signalling and ROS in the ovarian niche. Finally, a previous report demonstrated that Mam does not act in the conventional Notch-transduction pathway in the response of FSCs to the Hedgehog ligand in the *Drosophila* ovary [58]. These authors also showed that overexpression of a dominant-negative, truncated Mam protein lacking an acidic cluster involved in transcriptional activation could interfere with Notch signalling in the ovary but did not affect Mam$^{Long}$ activity in FSCs, where *mam* acts in a Notch-independent fashion. In our work, we demonstrate that high *mam* levels and the overactivation of the Notch pathway have distinct outcomes, supporting our claim that the naturally occurring upregulation of *mam* mRNA levels acts independently of Notch signalling. Furthermore, while the nature of the partners in niche cells containing elevated Mam amounts is currently unknown, they are most likely not the canonical effectors of the Notch pathway Su(H) and $N^{intra}$, suggesting that full-length Mam is able to interact with other partners to regulate GSC numbers during ageing. We envision a situation in which increased levels of the Mam co-activator could be acting in the niche to integrate multiple signalling pathways that ultimately lead to niche ageing. The consequences of this regulatory cross-talk are the reduction in niche cell–stem cell adhesion and niche signals such as Hedgehog and the increase in oxidative stress characteristic of aged niche cells.

## 3.3. Defective *D*E-cadherin trafficking in aged niches

A landmark of aged ovarian niches is a characteristic reduction in the cadherin-mediated adhesion between stem cells and the supporting CpCs. In addition to confirming that ageing and *mam* overexpression reduce *D*E-cadherin amounts, we report that the decreased *D*E-cadherin levels in aged niches could be due to a less effective trafficking and/or to a differential balance between recycling and endocycling for destruction via the lysosomal compartment. This idea is supported both by the FRAP experiments—which indicate an age-dependent decline in recycling capacities of the CpCs—and by the decrease in Rab11 (also observed in *mam*-overexpressing niches) and in Nuf contents in aged cells, which could be indicative of unbalanced *D*E-cadherin recycling/degradation. Together with the finding that maturing niches present a reduced capacity to replace the fusion protein Resil::GFP, our results thus point towards a general decay in membrane protein turnover with age, hence implicating ageing in the correct trafficking of membrane-associated proteins. Whether this effect is of general importance and whether it is due to the increasing levels of oxidative stress in old niche cells

remains to be seen, but the fact that protein replacement dynamics are related to ageing and that *D*E-cadherin recycling seems more sensitive to ageing than a control protein is of particular interest to the field, in view of the importance of stem cell–niche cell adhesion for tissue homeostasis [59,60].

# 4. Material and Methods

The following is a short version of the material and methods used in this study. A complete account can be found in the electronic supplementary material.

## 4.1. Germline stem cell quantifications

GSCs were identified by their position within the germarium (they are located just underneath the CpCs), by their size (they are larger that differentiating germline cells) and by the presence of a spectrosome in their cytoplasm. GSC spectrosomes adopt different morphologies depending on the cell-cycle phase. The number of GSCs found in a niche shows some variability, even in flies of the same genotype and raised in the same vial, and it ranges from 2 to 3 GSCs/germarium in the different control females used in this study. Because of this variability, in all of the experiments where we scored GSC numbers, we analysed control flies in parallel with their experimental siblings. To normalize the changes observed, we show the variation in the percentage of GSCs/germarium in all pertinent graphs. Thus, comparisons should be made between control and experimental niches of the same experiment and not between genotypes of different treatments and/or conditions. All quantifications were obtained from at least five different females/replicate of a given phenotype and from a minimum of two biological replicates.

We also performed a number of controls to make sure that the chromosomes used in this study did not affect GSC numbers specifically. Thus, we quantified GSCs in females bearing the TM6B balancer or chromosomes carrying the *bab1-Gal4*, *tub- G80$^{ts}$*, *ptc-Gal4* or *hh-Gal4* constructs and determined that they did not have a significant effect on the GSC pool present in the niches.

## 4.2. Fly stocks

Stocks obtained from public repositories were: *mam-Gal4* (BDSC cat. no. 49444, RRID:BDSC_49444), *hh-Gal4* [61] (DGGR cat. no. 118117, RRID:DGGR_118117), *ptc-Gal4* (BDSC cat. no. 45900, RRID:BDSC_45900), *D*E-cadherin::GFP (BDSC cat. no. 60584, RRID:BDSC_60584), *UAS-mam* (long isoform; BDSC cat. no. 27743, RRID:BDSC_27743), *UAS-mam*$^{Short}$ [28], *UAS-N*$^{intra}$ (BDSC cat. no. 52008, RRID:BDSC_52008), *UAS-SOD1* (BDSC cat. no. 33605, RRID:BDSC_33605), *tub-Gal80$^{ts}$* (BDSC cat. no. 7017, RRID:BDSC_7017), *UAS-mam RNAi* (BDSC cat. no. 28046, RRID:BDSC_28046), *UAS-Delta RNAi* (BDSC cat. no. 34938, RRID:BDSC_34938) and *UAS-Dicer* (BDSC cat. no. 24650, RRID:BDSC_24650). Other stocks used were: *Resille::GFP* [62], *UAS-DE-cadherin* [63], *bab1-Gal4* [27] and *traffic-jam-Gal4* [64]. To control the expression of different transgenes in adult niches, we used the *tub-Gal80$^{ts}$* system. Placing the flies at 29°C would induce Gal4-mediated gene expression, whereas keeping the adults at 18°C blocks Gal4-induced gene expression.

royalsocietypublishing.org/journal/rsob    Open Biol. 9: 190127

## 4.3. Immunohistochemistry

Primary antibodies used were the following: mouse anti-Hts (1B1) (DSHB cat. no. 1b1, RRID:AB_528070), 1 : 100; rabbit anti-Vasa (a gift from R. Lehmann), 1 : 3000; goat anti-GFP, FITC conjugated (Abcam cat. no. ab6662, RRID:AB_305635), 1 : 500; alpaca anti-GFP-Booster_Atto488 (ChromoTek cat. no. gba488-100, RRID:AB_2631386), 1 : 200; mouse anti–Lamin C (DSHB cat. no. lc28.26, RRID:AB_528339), 1 : 30; rat anti-DE-cadherin (DSHB cat. no. DCAD2, RRID:AB_528120), 1 : 50; guinea pig anti-Hh (this work; raised against a His-tagged fragment of the Hh protein (aa. 82-257) produced in *Escherichia coli*; polyclonal antibodies were affinity purified), 1 : 100; rabbit anti-phospho-Mad (Abcam cat. no. ab52903, RRID: AB_882596), 1 : 1000; rabbit anti-cleaved Dcp-1 (Cell Signaling Technology cat. no. 9578, RRID:AB_2721060), 1 : 100; guinea pig anti-Nuf [33], 1 : 500; mouse anti-Rab11 (BD Biosciences cat. no. 610657, RRID:AB_397984), 1 : 100; mouse anti-Patched (DSHB cat. no. Drosophila Ptc (Apa 1), RRID:AB_528441), 1/ 100. DNA dye: Hoechst (Sigma B2883 10 mg ml$^{-1}$ in H$_2$O).

## 4.4. Image analysis

Confocal images of control and experimental samples processed in parallel were obtained with exactly the same settings. Average fluorescence intensity was quantified with ImageJ. See electronic supplementary material and [65] for further details.

## 4.5. Fluorescence recovery after photobleaching

Regions of interest per niche were bleached with the 488 nm and 405 nm laser lines at 100% power for three 2.6 s long scans (400 Hz). Three Z-stacks were then collected at 10 min time intervals.

## 4.6. Isolation of niche cells and microarray analysis

Preparation of single-cell suspensions was done by optimizing a previous protocol [66]. Niche cells were selected by FACS according to size, viability and GFP-signal intensity, which allowed TFCs and CpCs to be separated from ECs. RNA isolated from three biological replicates for each of the three time points was used to prepare amplified cDNA libraries. The nine cDNAs were hybridized to GeneChip *Drosophila* Genome 2.0 Arrays from Affymetrix.

## 4.7. Quantification of mRNA levels by qPCR and ddPCR

The relative mRNA amounts of *mam* long and short isoforms were determined by real-time quantitative PCR (qPCR; CFX Connect; Bio-Rad) using the comparative cycle threshold ($C_T$) method [67]. Pico-profiled samples from sorted niche cells (TFCs and CpCs; see above) were used to quantify mRNA levels of *mam* long and short isoforms by Droplet Digital PCR (ddPCR).

## 4.8. Statistical analysis

Samples were collected from at least five different adult females. Graphs represent the mean values and standard errors. Significant differences were determined using two-tailed Student's *t*-tests. In all figures, only *p*-values of two-tailed *t*-tests considered statistically significant ($p \leq 0.05$) are indicated. Thus, the absence of asterisks indicates no statistical significance between control and experimental values.

Data accessibility. The article's supporting data and materials are referred to in the manuscript. The electronic supplementary material contains data relevant to our discoveries.

Authors' contributions. M.L.-P. carried out experimental work, participated in data analysis and helped with the writing of the manuscript; M.M.-M. performed experiments and analysed data; A.G.-R. conceived, designed and coordinated the study, wrote the manuscript, performed experiments, analysed data and provided financial support. All authors gave final approval for publication and agree to be held accountable for the work performed therein.

Competing interests. We declare we have no competing interests.

Funding. This work was funded by the Spanish State Agency for Research (MCUI/AEI; grant nos. BFU2015-65372-P, PGC2018-097115-B-I00 to A.G.-R. and MDM-2016-0687) and by the European Regional Development Fund (http://ec.europa.eu/regional_policy/en/funding/erdf/).

Acknowledgements. We thank the BDSC for fly stocks and the DSHB (University of Iowa) and S. Sotillos for antibodies. The technical help of A. García, C. Díaz, K. García, A. Fernández-Miñán and A. E. Rosales is appreciated. We are grateful to M. J. Sánchez for her guidance in the cell-sorting procedures and to M. D. Martín-Bermudo, A. E. Rosales, D. Even-Ros, G. Villa-Fombuena, S. Sotillos and M. J. Sánchez for comments on the manuscript.

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
