## [Reviewer comments · Open Biology]

Review History

RSOB-19-0127.R0 (Original submission)

Review form: Reviewer 1

Recommendation

Accept with minor revision (please list in comments)

Do you have any ethical concerns with this paper?

No

Comments to the Author

This is a very good work that deals with the important issue of aging, and how aging impacts on the number of stem cells. There is very little known about this topic, and even less about how changes in the stem cells niche environment during aging affect stemcellness.

I only have two major comments. The first one is regarding the mam RNAi. The authors do a great job in characterising aging cells, as well as recapitulating some of this data in young cells that overexpress mam. Now, they also try to reduce mam levels in aged cells, but face some

complex results. However, I still think the authors could explore this further. For example, would it be possible to analyze the 4weeks at 22degrees mam RNAi situation a bit more? At 22 degrees, you can see that the number of GSCs at 4 weeks mam RNAI does not seem to drop, compared to controls, is this correct? so, in this situation, what about cadh, ROS, etc...?

The other comment is that I would like to see a better characterization of ROS and SOD1 in aging females, which I think is missing in this work, specially if they could link it to what happens when mam levels are reduced (following my comment above). Also, the authors state that mam overexpression in niche cells increases ROS levels, how do they think this may be happening? I missed some discussion on this point.

Minor comments:

Figure 4 title states that 'DE-cadherin recycling and Rab11 levels change during ageing or upon mam overexpression', but I think this has not been studied for Cadh in the case of mam overexpression.

Ptc in green in figure5E, which is labeled like that in the inset, seems not to be present in the image

Review form: Reviewer 2

Recommendation

Major revision is needed (please make suggestions in comments)

Do you have any ethical concerns with this paper?

No

Comments to the Author

Please find my comments and suggestions in the attached file.

Decision letter (RSOB-19-0127.R0)

24-Jul-2019

Dear Professor González-Reyes,

We are writing to inform you that the Editor has reached a decision on your manuscript RSOB-19-0127 entitled "mastermind regulates niche ageing independently of the Notch pathway in the *Drosophila* ovary", submitted to Open Biology.

As you will see from the reviewers' comments below, there are a number of criticisms that prevent us from accepting your manuscript at this stage. The reviewers suggest, however, that a revised version could be acceptable, if you are able to address their concerns. If you think that you can deal satisfactorily with the reviewer's suggestions, we would be pleased to consider a revised manuscript.

The revision will be re-reviewed, where possible, by the original referees. As such, please submit the revised version of your manuscript within six weeks. If you do not think you will be able to meet this date please let us know immediately.

When submitting your revised manuscript, please respond to the comments made by the referee(s) and upload a file "Response to Referees" in "Section 6 - File Upload". You can use this to document any changes you make to the original manuscript. In order to expedite the processing of the revised manuscript, please be as specific as possible in your response to the referee(s).

Please see our detailed instructions for revision requirements
<https://royalsociety.org/journals/authors/author-guidelines/>

Sincerely,
The Open Biology Team
mailto: openbiology@royalsociety.org

Editor's Comments to Author(s):
Please address comments of reviewers or otherwise give an argued response

Handling Editor's response:
Both Reviewers agree that the ms is overall of sufficient interest, and is in principle of acceptable quality for publication in Open Biology. The Reviewers further agree that it would be important to investigate in more detail what are the phenotypes in aged flies compared to young flies, and how mam overexpression and RNAi affect DE-Cadherin and ROS.

Reviewer(s)' Comments to Author(s):

Referee: 1

Comments to the Author(s)
This is a very good work that deals with the important issue of aging, and how aging impacts on the number of stem cells. There is very little known about this topic, and even less about how changes in the stem cells niche environment during aging affect stemcellness.

I only have two major comments. The first one is regarding the mam RNAi. The authors do a great job in characterising aging cells, as well as recapitulating some of this data in young cells that overexpress mam. Now, they also try to reduce mam levels in aged cells, but face some complex results. However, I still think the authors could explore this further. For example, would

it be possible to analyze the 4weeks at 22degrees mam RNAi situation a bit more? At 22 degrees, you can see that the number of GSCs at 4 weeks mam RNAI does not seem to drop, compared to controls, is this correct? so, in this situation, what about cadh, ROS, etc...?

The other comment is that I would like to see a better characterization of ROS and SOD1 in aging females, which I think is missing in this work, specially if they could link it to what happens when mam levels are reduced (following my comment above). Also, the authors state that mam overexpression in niche cells increases ROS levels, how do they think this may be happening? I missed some discussion on this point.

Minor comments:

Figure 4 title states that 'DE-cadherin recycling and Rab11 levels change during ageing or upon mam overexpression', but I think this has not been studied for Cadh in the case of mam overexpression.

Ptc in green in figure5E, which is labeled like that in the inset, seems not to be present in the image

Referee: 2

Comments to the Author(s)

Please find my comments and suggestions in the attached file.

Author's Response to Decision Letter for (RSOB-19-0127.R0)

See Appendix A.

RSOB-19-0127.R1 (Revision)

Review form: Reviewer 1

Recommendation

Accept as is

Do you have any ethical concerns with this paper?

No

Comments to the Author

The authors have cleared all my concerns and I recommned the manuscript to be published as it is.

Review form: Reviewer 2

Recommendation

Accept with minor revision (please list in comments)

Do you have any ethical concerns with this paper?

No

Comments to the Author

Overall, the authors have addressed most of my original concerns with the manuscript. One remaining issue is in organization of the data/clarity in description of phenotypes. This is particularly true for the Notch dependent vs independent roles for mam in the niche. Data in the manuscript suggests that the increased mam expression with age induces GSC loss in a Notch-independent manner. By contrast, the authors suggest in their response to reviewers that the mam RNAi phenotype of GSC loss is likely the result of a Notch-dependent effect. While I do not believe additional experiments are necessary to test this, a statement to this effect should be added to the results. In addition, the manuscript would benefit from presenting all Notch-related data (including the test of mamSHORT vs mamLONG isoforms) into a single section rather than spread throughout the results.

Decision letter (RSOB-19-0127.R1)

07-Oct-2019

Dear Professor González-Reyes,

We are pleased to inform you that your manuscript RSOB-19-0127.R1 entitled "mastermind regulates niche ageing independently of the Notch pathway in the Drosophila ovary" has been accepted by the Editor for publication in Open Biology. The reviewer(s) have recommended publication, but also suggest some minor revisions to your manuscript. Therefore, we invite you to respond to the reviewer(s)' comments and revise your manuscript.

Please submit the revised version of your manuscript within 7 days. If you do not think you will be able to meet this date please let us know immediately and we can extend this deadline for you.

1) A text file of the manuscript (doc, txt, rtf or tex), including the references, tables (including captions) and figure captions. Please remove any tracked changes from the text before submission. PDF files are not an accepted format for the "Main Document".

2) A separate electronic file of each figure (tiff, EPS or print-quality PDF preferred). The format should be produced directly from original creation package, or original software format. Please note that PowerPoint files are not accepted.

3) Electronic supplementary material: this should be contained in a separate file from the main text and meet our ESM criteria (see <http://royalsocietypublishing.org/instructions-authors#question5>). All supplementary materials accompanying an accepted article will be treated as in their final form. They will be published alongside the paper on the journal website and posted on the online figshare repository. Files on figshare will be made available approximately one week before the accompanying article so that the supplementary material can be attributed a unique DOI.

Online supplementary material will also carry the title and description provided during submission, so please ensure these are accurate and informative. Note that the Royal Society will not edit or typeset supplementary material and it will be hosted as provided. Please ensure that the supplementary material includes the paper details (authors, title, journal name, article DOI). Your article DOI will be 10.1098/rsob.2016[*last 4 digits of e.g. 10.1098/rsob.20160049*].

4) A media summary: a short non-technical summary (up to 100 words) of the key findings/importance of your manuscript. Please try to write in simple English, avoid jargon, explain the importance of the topic, outline the main implications and describe why this topic is newsworthy.

Data-Sharing

It is a condition of publication that data supporting your paper are made available. Data should be made available either in the electronic supplementary material or through an appropriate repository. Details of how to access data should be included in your paper. Please see <http://royalsocietypublishing.org/site/authors/policy.xhtml#question6> for more details.

Data accessibility section

Sincerely,

The Open Biology Team
<mailto:openbiology@royalsociety.org>

Reviewer(s)' Comments to Author:

Referee: 2

Comments to the Author(s)

Overall, the authors have addressed most of my original concerns with the manuscript. One remaining issue is in organization of the data/clarity in description of phenotypes. This is particularly true for the Notch dependent vs independent roles for mam in the niche. Data in the manuscript suggests that the increased mam expression with age induces GSC loss in a Notch-independent manner. By contrast, the authors suggest in their response to reviewers that the mam RNAi phenotype of GSC loss is likely the result of a Notch-dependent effect. While I do not believe additional experiments are necessary to test this, a statement to this effect should be added to the results. In addition, the manuscript would benefit from presenting all Notch-related data (including the test of mamSHORT vs mamLONG isoforms) into a single section rather than spread throughout the results.

Referee: 1

Comments to the Author(s)

The authors have cleared all my concerns and I recommend the manuscript to be published as it is.

Author's Response to Decision Letter for (RSOB-19-0127.R1)

See Appendix B.

Decision letter (RSOB-19-0127.R2)

21-Oct-2019

Dear Professor González-Reyes,

We are pleased to inform you that your manuscript entitled "mastermind regulates niche ageing independently of the Notch pathway in the *Drosophila* ovary" has been accepted by the Editor for publication in *Open Biology*.

Article processing charge

Please note that the article processing charge is immediately payable. A separate email will be sent out shortly to confirm the charge due. The preferred payment method is by credit card; however, other payment options are available.

Sincerely,

The Open Biology Team
mailto: openbiology@royalsociety.org

Appendix A

Reviewer 1 RSOB-19-0127 - Comments to the Author(s)

This is a very good work that deals with the important issue of aging, and how aging even less about how changes in the stem cells niche environment during aging affect stemcellness.

We thank the reviewer for their kind words and for their effort in reviewing the MS.

I only have two major comments. The first one is regarding the mam RNAi. The authors do a great job in characterising aging cells, as well as recapitulating some of this data in young cells that overexpress mam. Now, they also try to reduce mam levels in aged cells, but face some complex results. However, I still think the authors could explore this further. For example, would it be possible to analyze the 4weeks at 22degrees mam RNAi situation a bit more? At 22 degrees, you can see that the number of GSCs at 4 weeks mam RNAi does not seem to drop, compared to controls, is this correct? so, in this situation, what about cadh, ROS, etc...?

The reviewer is right: *bab1>UAS-mam RNAi* flies grown at 22°C for 4 weeks behave as controls when it comes to GSC numbers. Following the reviewer's suggestion, we have now looked at *DE-cad* and *Hh* levels in their GSC niches and found that there are no differences with the respective controls (see new Fig S2B). As for ROS activity, we could not study the state of ROS in this condition, as our only test (see below) to check whether the oxidative stress in niche cells is affected is trying to rescue GSC numbers by *SOD1* overexpression, an approach that is not suitable in this context since GSC numbers are not different in control and experimental flies.

The other comment is that I would like to see a better characterization of ROS and *SOD1* in aging females, which I think is missing in this work, specially if they could link it to what happens when mam levels are reduced (following my comment above). Also, the authors state that mam overexpression in niche cells increases ROS levels, how do they think this may be happening? I missed some discussion on this point.

In one of our experiments, we showed that *SOD1* overexpression in niche cells does not seem to affect GSC numbers, as *bab1>sod1* kept for 2 weeks at 29°C behaved like controls. From this and from the fact that the GSC loss typical of aged niches can be rescued by *SOD1* ectopic expression, we would conclude that it is the aged-induced increase in oxidative stress what affects niche cell activity and thus GSC maintenance. As for characterizing ROS in ageing females. We have tried to measure ROS levels directly, by means of a fluorometric assay utilising the CM-H2DCFDA ROS-species detector to quantify ROS amounts in ovaries (Manta et al, Fly 2017). Unfortunately, while we could detect clear, consistent signals in later stages, germaria (and cap cells in particular) were very difficult to measure reliably due to their small size and the diffuse staining given by the chemical. Thus, we resorted to using the *SOD1* overexpression rescue as a proxy to define whether the production of ROS was an issue in *mam*-induced niche ageing. Our data indicate that *SOD1* overexpression rescues the GSC loss induced by *mam* overexpression.

Lastly, regarding the question of why would higher *mam* levels induce oxidative stress -most likely by increasing ROS levels-, we cannot give a clear explanation here. We can only speculate that overexpression of *mam* for 2 weeks induces an accumulation of ROS-dependent damage to the cell which ultimately results in impaired signalling, recycling and adhesion in the niche. Perhaps high *mam* amounts

accelerate the cell's metabolism thus increasing the generation of free radicals to the extent that the cell cannot cope with them efficiently. Alternatively, or in addition, *mam* overexpression might impede the normal activity of the antioxidant defense system present in niche cells. We have modified the Discussion (lines 390 to 403) to address this point.

Minor comments

Figure 4 title states that 'DE-cadherin recycling and Rab11 levels change during ageing or upon *mam* overexpression', but I think this has not been studied for Cadh in the case of *mam* overexpression.

The reviewer is right. The figure legend has been corrected. Now it reads “*DE*-cadherin recycling and Rab11 levels change during ageing. *mam* overexpression affects Rab11 amounts”.

Ptc in green in figure5E, which is labeled like that in the inset, seems not to be present in the image

The green channel is indeed present in the inset, even though it is nearly undetectable as most of the green dots are hidden by the DNA (blue) and Vasa (red) stainings. This is why we decided to show an enlarged panel of the Ptc signal on its own.

Reviewer 2 RSOB-19-0127 - Comments to the Author(s)

Lobo-Pecellin et al. present compelling data linking increased expression of mastermind in the somatic niche of aged germaria to multiple, common defects associated with stem cell and niche aging. The authors find through microarray that *mam* expression is increased in niche cells from aged flies and that overexpression of *mam* in the somatic niche induces precocious aging, denoted by loss of GSCs. Although a known regulator of Notch signaling, *Mam* appears to control aging of the somatic niche in a Notch-independent manner in the germarium. Instead, the authors find that *mam* overexpression caused decreased levels of DE-Cadherin, decreased expression of Hh and increased presence of reactive oxygen species. Importantly, only overexpression of an antioxidant is capable of rescuing GSC loss due to increased *mam* expression, indicating that increased ROS may play a central role in niche aging. Overall, the manuscript provides a number of interesting insights into the process of niche aging with some associated mechanism. While there are some minor issues, the major need for revision is to increase clarity in discussion of the phenotypes associated with *mam* overexpression in young flies versus defects in aged flies and how/if those phenotypes relate to one another.

We appreciate the reviewer's words regarding our work and thank them for their thorough appraisal of our manuscript. As for modifying the presentation of the phenotypes and how they relate to each other and to physiological ageing, we have made some changes to the Discussion that hopefully provide a clearer read of the MS. See below for details.

Major Concerns

The authors find in *bab1>mam* germaria:

Decreased DE-Cadherin levels

Decreased Rab11 levels
Decreased Hh expression
Increased ROS
In aged flies they find:
Decreased DE-Cadherin levels
Decreased Rab11 levels
Decreased recycling of membrane components, including DE-Cadherin

The connection between these phenotypes and the degree to which *mam* overexpression recapitulates the aging phenotypes are not made completely clear. In addition, the relative importance/significance of each of these phenotypes is not discussed. The manuscript would be much stronger if reorganized and rewritten to highlight the truly significant findings (*mam* expression increased with age, overexpression causing GSC loss in young flies, rescue of *mam* overexpression with antioxidant expression), to discuss the new factors contributing to aging identified in this study (decreased membrane trafficking) and to remark about but not focus on the phenotypes of *mam* overexpression that do not seem to have a direct functional output in terms of aging (decreased Hh expression). The way the manuscript is currently written, each of those phenotypes is given the same relative importance yet there are obvious differences in terms of the relevance of each of these defects to niche aging.

We have modified the Discussion section to reflect the points raised by the reviewer. Thus, we have now included a new fragment on the oxidative stress induced by *mam* overexpression (see also our response to Reviewer 1) and have emphasized the correlation found between ageing and impaired trafficking of membrane proteins. Here, however, we had to be cautious, as we have not proven the causal relationship between ageing and recycling. We have also clarified the importance of those factors that do have a direct impact on GSC numbers in the niche and that are the novel findings of our contribution.

In particular:

Overexpression of both *mam* and the antioxidant SOD1 rescues the GSC loss associated with increased *mam*. While this would seem to suggest that increased ROS is the main contributor to precocious aging upon *mam* overexpression, this data is given equal weight of discussion as other phenotypes. In addition, the cause-effect relationship of increased ROS with the other defects observed upon *mam* overexpression are not described. Does SOD1/*mam* co-expression rescue DE-Cadherin levels? Hh levels? The manuscript would be stronger with results from experiments addressing these questions included.

In our original experiment, we only looked at GSC numbers to check if high *mam* levels caused oxidative stress that could be rescued with SOD1 overexpression. In accordance with the reviewer's comment, we have now repeated the cross and have quantified DE-cad and Hh levels to find that control (*UAS-sod1 + UAS-mam* without *bab1-Gal4*) and experimental females (*bab1>mam + sod1*) contain similar amounts of DE-cad and Hh. These data have been added to Suppl. Fig S7 (new panels S7A and S7B) and mentioned in the text.

Does SOD1 expression rescue age-related decline in GSC numbers as well as in the mam overexpression?

This experiment we did not do, as it had been published previously that SOD1 overexpression in niche cells utilising the *hh-Gal4* line rescued GSC decline in aged germaria (Pan et al. Cell Stem Cell 2007).

Pan et al., 2007 show that increased DE-Cadherin expression in aged germaria can rescue GSC loss. Yet these authors find that overexpression of DE-Cadherin does not rescue mam overexpression defects. A more thorough discussion of this discrepancy and potential explanations should be included in the manuscript.

We have read carefully the Pan et al. publication and can confirm they show that DE-cad overexpression in the germline (*nos>FLAG-shg*) prevents age-dependent GSC loss. However, they could not induce ectopic DE-cad expression in niche cells as they claim that “*Unfortunately, hh-gal4-driven expression of E-cadherin leads to death of young adults, precluding us from directly testing the idea of whether an increase in expression of E-cadherin in the aged niche could also stabilize GSCs (data not shown)*”. Thus, the extent to which overexpression of DE-cad in niche cells can rescue age-dependent GSC loss requires further experimentation.

The authors describe reduced membrane trafficking in the aged somatic niche, evidenced by diminished recovery of DE-Cadherin and Resil::GFP in FRAP experiments. The interpretation is that diminished DE-Cadherin trafficking contributes to the reduced adhesion of GSCs to the niche and, thus, GSC loss. However, they also observed diminished DE-Cadherin with mam overexpression yet did not assay for altered trafficking. It would be important to differentiate between the possibilities that altered trafficking in aged flies is due to increased mam expression (in which case the young flies with mam overexpression should also have altered trafficking) or if the trafficking defect is independent of mam expression (in which case young flies with mam overexpression should have normal trafficking).

We report in the MS that aged niches show impaired Resil::GFP and DE-cad::GFP recovery in FRAP experiments, that Rab11 and Nuf levels are reduced in aged germaria and that *mam* overexpression (2 weeks at 29 °C) diminished Rab11 levels. However, we could detect neither a change in Nuf levels in *bab1>mam* (2 weeks at 29 °C) nor a different behaviour to controls with respect to DE-cad::GFP recovery in FRAP experiments in *bab1>mam* (1 week at 29 °C; we could not do 2 weeks as the DE-cad::GFP signal was too low to perform the FRAP experiments). From these data we concluded that aged niches reduce membrane protein recycling and that ageing favours the sorting of internalized DE-cad towards the lysosomal pathway.

The question posed by the reviewer is relevant and is one we sought to answer during the course of our experiments. This is the reason why we looked at Rab11 and Nuf levels and why we tried FRAP experiments in *bab1>mam* niches. Unfortunately, and as mentioned above, *bab1>mam* niches 1 week at 29°C did not give significant fluorescent recovery differences and 2 weeks at 29°C we could not analyse. Thus, we are afraid we cannot assess the role of increased *mam* levels in the reduced membrane-protein recycling reported for physiologically-aged niches.

The authors observe GSC loss upon expression of *mam* RNAi in the somatic niche. The conclusion is that precise levels of *mam* are critical to maintain proper niche function and GSC maintenance. However, no experiments were done to determine if the cause of GSC loss in the RNAi condition is the same as with *mam* overexpression (ie, decreased DE-Cadherin, increased ROS).

The reviewer is correct. Because *mam* expression increases in 3- and 4-week old niche cells and this is accompanied by significant GSC loss, we wished to test if a partial reduction in *mam* levels would rescue the decrease in GSC numbers. To this end, we grew *bab1>mam* RNAi flies at 22°C or at 25°C for 4 weeks. Flies kept at 22°C behaved as controls, while flies grown at 25°C induced GSC loss. This could be interpreted as *mam* levels being reduced too little at 22°C to allow for a GSC rescue. As for the phenotype of *bab1>mam* RNAi flies kept at 25°C for 4 weeks, we believe it reflects the fact that *mam* also acts in the canonical Notch signalling pathway. This is based on our own findings that demonstrate that reduced Notch signalling in niche cells induced GSC loss (see below for a more detail explanation of our experiments regarding the Notch pathway). Since the activity of this pathway has been extensively studied in previous publications (see for instance Hsu + Drummond-Barbosa PNAS 2009 and Tseng et al. PLoS G 2014) we hope the reviewer agrees that their suggested experiments fall outside of the scope of our work.

Relatedly, *mamshort* acts as a dominant negative to *mamlong* function. So overexpression of *mamshort* should presumably alter the balance of *mamlong* activity. If this balance is critical (such that RNAi against *mam* or overexpression both cause GSC loss) then why do the authors not observe a GSC loss phenotype upon *mamshort* overexpression?

This question we have asked ourselves a number of occasions but have not been able to find a definitive answer to it. It has been published that *mam^{short}* overexpression in other tissues mimics Notch pathway loss of function. We have shown that overexpressing *mam* RNAi (*bab1>mam* RNAi, 1 week at 29°C) does reduce GSC numbers in the niche. However, a similar treatment but in this occasion overexpressing *mam^{short}* (*bab1>mam^{short}*, 1 week at 29°C) does not lead to a significant decrease in GSCs. In absence of a clear explanation for these findings, we can only speculate that perhaps the extent to which *mam* function is abolished in niche cells in the RNAi experiment is higher than in the *mam^{short}* approach. This could be due to different dynamics of the Su(H)/N intra/Mam ternary complex in niche cells compared to other cell types in which overexpression of the short isoform recapitulates Notch loss of function phenotypes.

Finally, the authors show that the *mam* overexpression phenotype is Notch-independent. It would be interesting to know if the *mam* RNAi phenotype IS Notch dependent.

We agree with the reviewer that this is an interesting point. One of the main results of our work is that high *mam* levels in niche cells induces a decrease in GSC numbers and that this is independent of the Notch pathway. In addition, Tseng et al. PLoS G 2014 published that GSC loss in ageing niches correlates with decreased Notch signalling in niche cells, indicating that diminished Notch activity in the niche might induce a reduction in GSC numbers.

As mentioned by the reviewer, we show that *bab1>mam RNAi* ovaries contain fewer GSCs than controls. Considering that *mam* acts in the canonical transduction of the Notch signal, our finding agrees with the results by Tseng et al. To prove further that reduced Notch signalling in the niche cells induces GSC loss, we confirmed that *bab1>DI RNAi* ovaries hosted fewer GSCs than controls. This reasoning is mentioned in the text, which reads “*it (Notch signalling) decreases in 7-week old niche cells [16, 20], suggesting that low levels of Notch pathway activity in niche cells correlate with the GSC loss characteristic of aged niches, in line with our finding that bab1>mam RNAi females grown at 25°C contained fewer GSCs than their controls. [...] We confirmed that loss of function of the Notch pathway (by reducing the amount of the Delta ligand in niche cells) reduced GSC numbers [...], corroborating that adult germlaria depended on Notch signalling in niche cells for GSC maintenance*”.

Thus, we have shown that reduced Notch signalling in niche cells correlates with GSC loss and that decreasing *mam* levels gives the same phenotype. Considering the role of *mam* in Notch signal transduction, we believe this evidence supports our suggestion that Notch signalling (most likely mediated by *mam*) regulates GSC numbers.

Minor points

When mastermind is first introduced it would be helpful to have a description of what the protein is/its function.

Done

The description of *bab1-Gal4* as expressing in all cells of the somatic niche should be moved earlier in the results when the Gal4 is first used.

Done

In the section regarding trafficking of DE-Cadherin the authors conclude that aged niches favor DE-Cad sorting to the lysosomal degradation pathway rather than recycling. While there is evidence suggesting a decrease in recycling there is no direct evidence that DE-Cadherin instead is being redirected to the lysosome. The authors should either soften the language here or show co-localization of internalized DE-Cadherin with lysosomal markers.

Done. It now reads “*our results may suggest that ageing favours the sorting of internalized DE-cadherin (and likely many other membrane proteins such as Resil) towards the lysosomal degradation pathway rather than to the Rab11-mediated recycling*”.

The results section title “*mam* expression regulates signaling in the ovarian niche” is a bit misleading. While the authors find that Hh expression is decreased the actual “signaling” (ie, pathway activation in responding cells) remains unchanged. In fact, this section was a bit challenging to understand and reconcile. The manuscript might be better served by simply removing the Hh data, as it’s not clear what relevance it has to the overall *mam* overexpression/aging phenotypes.

The reviewer is right. What we find is a reduction in Hh levels in niche cells, but this does not seem to have an effect on the activation of the pathway in the target cells.

The title of the section now reads “*mam expression regulates Hedgehog levels in the ovarian niche*”.

We have decided to leave the Hh data in as it is the first report that indicates that ageing (and *mam* overexpression) affects Hh amounts. While we do not know the reason(s) as to why lower Hh levels do not affect (allegedly) *hh* signalling, this in itself is an interesting finding.

In the discussion, the authors enumerate a number of Notch-independent roles for Mam, one of which is in regulation of Wnt signaling. The Xie lab has shown that Wnt signaling regulates the redox state of niche cells and GSCs in both the testis and germarium (Wang et al, 2015). Given the connection with ROS in this manuscript, this paper should at least be mentioned in the discussion to provide a compelling link between Mam, Wnt and ROS.

Done. We have modified the Discussion and added a sentence citing this work. Thanks for the suggestion.

Appendix B

Reviewer 2 RSOB-19-0127.R1 - Comments to the Author(s)

Overall, the authors have addressed most of my original concerns with the manuscript. One remaining issue is in organization of the data/clarity in description of phenotypes. This is particularly true for the Notch dependent vs independent roles for mam in the niche. Data in the manuscript suggests that the increased mam expression with age induces GSC loss in a Notch-independent manner. By contrast, the authors suggest in their response to reviewers that the mam RNAi phenotype of GSC loss is likely the result of a Notch-dependent effect. While I do not believe additional experiments are necessary to test this, a statement to this effect should be added to the results.

We have added a new sentence to this end (lines 135-136).

In addition, the manuscript would benefit from presenting all Notch-related data (including the test of mamSHORT vs mamLONG isoforms) into a single section rather than spread throughout the results.

I have gone through the text with care and I find it very difficult to reorganise the text so that the *mam^{Short}* and *mam^{Long}* can be included in the Notch section. Thus, we prefer to leave the organisation of the text as it is and hope that the reader can still follow our description of the results.